# Adaptive Fractional-Order Anti-Saturation Synchronous Control for Dual-Motor Systems

**Yongbin Zhong, Jian Gao *** and **Lanyu Zhang**

State Key Laboratory of Precision Electronic Manufacturing Technology and Equipment, Guangdong University of Technology, Guangzhou 510006, China
* Correspondence: gaojian@gdut.edu.cn; Tel.: +86-1356-0125-827

**Featured Application: High-speed synchronous motion system for precision electronic packaging equipment.**

**Abstract:** The synchronization error of a dual-motor system will seriously affect the motion profile accuracy. To solve this problem, an adaptive fractional-order anti-saturation synchronous control method based on fractional-order frequency-domain control theory is proposed in this paper. On the one hand, the proposed method performs a compensation on the closed-loop feedback control loop to unify the frequency-domain characteristics for a dual-motor system. With the frequency-domain characteristics' unification module, the dual-motor system will have the same response performance regarding the input signal. On the other hand, considering that the nonlinear problem of control voltage saturation will also cause the asynchronization problem of the dual-motor system, the proposed method involves an adaptive fractional-order anti-saturation module to prevent voltage saturation and eliminate the nonlinear effects. The experimental results verify that the proposed method can accurately avoid the saturation effect and effectively reduce the synchronization error of the dual-motor system, with a root-mean-square synchronization error reduction of 80.974%. Hence, the proposed method provides an effective solution for the high-precision synchronous motion of a dual-motor system.

**Keywords:** dual-motor system; synchronous motion; fractional-order frequency-domain control theory; saturation effect

## 1. Introduction

With the development of power electronics technology and high-performance permanent magnet materials, the permanent magnet linear synchronous motor (PMLSM) has gained significant attention. Because its structure has no mechanical friction problem, a large thrust force, and small volume, the PMLSM is widely used in transportation systems, machine tools, precision positioning platforms, and other fields [1,2]. In addition, the market demand for chip semiconductors has increased in recent years, leading to the rapid development of the semiconductor manufacturing industry; therefore, the high-speed and high-acceleration motion platform built by the PMLSM has gradually become an essential part of the equipment [3]. However, the performance requirements are becoming increasingly stringent, and the single-motor driving method can no longer meet the high-acceleration requirement. To overcome the thrust force limit of the PMLSM and further improve the efficiency of semiconductor manufacturing equipment, a dual-motor driving method has been designed and widely used [4,5]. A redundant direct-drive gantry platform is a common application of the dual-motor driving method. The Y-degree-of-motion freedom is driven by two PMLSMs simultaneously, and the rigid connection between the PMLSMs ensures that the thrust force can act on the beam together, realizing high speed, high acceleration, and a large stroke motion [6].

A dual-actuator system can usually achieve a higher acceleration motion, but it also introduces the asynchronous motion problem, which will cause tension in the connection structure and decrease the service life or even seriously damage the mechanical structure [7]. Scholars have proposed a variety of control methods for the synchronous motion of a dual-motor system, including the series synchronous control method [8], parallel synchro-nous control method [9], and cross-coupling synchronous control method [10]. Because the cross-coupling synchronous control method can adjust the control voltage of each motor in real time according to the synchronization error, it has certain advantages in ensuring the synchronous motion of the dual-motor system [11,12]. The control structure of the cross-coupling synchronous control method is simple; therefore, the accuracy of the control parameter selection is the key factor that determines the synchronous motion performance of the dual-motor system. Some scholars improved the synchronous control method by considering cross-coupling, such as the variable-gain cross-coupling synchronous control method [13], fuzzy neural network cross-coupling synchronous control method [14], and self-tuning cross-coupling synchro-nous control method [15]. These methods can improve the real-time adjustment ability of the parameters of the cross-coupling controller, thereby ensuring the adaptability of the control parameters under different synchronization errors. However, accurately establishing the parameter adjustment law of the cross-coupling synchronous control method requires long-term pre-learning, and the control parameter adjustment algorithm has a high degree of complexity, making it difficult for practical applications. Considering that the characteristics of the PMLSM system can be obtained through system identification, several methods were studied, for example, a model-based feed-back-feedforward decoupling control method [16], model-based adaptive synchronous control method [17], robust immersion and invariance adaptive coordinated control method [18], adaptive thrust-allocation based synchronous control method [19], and other related methods [20–22]. The introduction of model information can reduce the design complexity of the parameter adjustment method and provide a basis for control parameter selection, thus effectively suppressing the dual-motor system's synchronization error. However, these synchronous control methods adjust the control voltage of the PMLSM through position compensation or force compensation when synchronization error is observed. Since there is a time delay, the methods will have difficulty eliminating the synchronization error completely.

Through investigation, we found that the synchronization error is caused by the characteristic difference in the dynamic response of the dual-motor system, and, because the fractional-order deferential operator can accurately describe the control frequency-domain characteristics required by the controlled object, there are several control methods based on fractional-order control theory [23–25]. In literature [26], the authors developed a feedforward control method for PMLSM based on the fractional-order control theory. Through the accurate description of the system's control frequency-domain characteristics, the fractional-order feedforward control method improved the PMLSM tracking accuracy effectively for a single-motor system. However, for a dual-motor system, besides the tracking accuracy of each motor, the synchronization accuracy of the two motors is the key issue and needs to be further studied. Therefore, considering the advantage of the fractional-order control theory, this paper proposes an adaptive fractional-order anti-saturation synchronous control method to improve the synchronization accuracy of the dual-motor system. Through compensation, we unified the frequency-domain characteristics of the motors in a dual-motor system to ensure they have the same dynamic response characteristics. With the design of the frequency-domain characteristics' unification module (FDC-UM), the dynamic response characteristics of the two motors were adjusted to be consistent. Then, the adaptive fractional-order anti-saturation module (AFOAM) was designed to avoid the saturation effect of the control voltage. By calculating the theoretical peak voltage, the parameters of the AFOAM were determined by the relationship between the parameter selection and the FDC adjustment of the fractional-order lead-lag controller (FOLLC). Through the AFOAM, the dual-motor system can prevent the control voltage

from reaching the voltage limitation and thus guarantee the suppression of the FDC-UM for the synchronization error. Therefore, the proposed adaptive fractional-order anti-saturation synchronous method can actively suppress the generation of the synchronization error and avoid the saturation effect, ensuring a dual-motor system with high-precision synchronous motion performance.

The remainder of this paper is organized as follows. Section 2 introduces the kinetic model of the dual-motor system and the proposed adaptive fractional-order anti-saturation synchronous control method in detail. In Section 3, based on the established relationship between the FDC and the dynamic response characteristics of the motor systems, the FDC-UM and the AFOAM are designed and finalized with the required parameters. In Section 4, the experimental setup is described, and experimental work is performed to verify the effectiveness and superiority of the proposed method. Section 5 presents the discussion, and Section 6 presents the conclusions.

## 2. The Proposed Synchronous Control Method for the Dual-Motor System

### 2.1. Kinetic Model of the Dual-Motor System

A schematic of the redundant direct-drive gantry platform is shown in Figure 1. A cross beam was installed on two linear guides arranged in parallel and jointly driven by two PMLSMs. Motors $Y_1$ and $Y_2$ were rigidly connected to the beam, and the stators were installed on the base where the two linear guides were located. For the sake of simplicity, this study only focused on the synchronous control of the *Y*-axis, and it is assumed that the motor X is rigidly connected to the cross-beam. Therefore, from Figure 1 and the dynamic analysis in literature [27], the kinetic model of the dual-motor system can be obtained as follows:

$$\begin{cases} M_1\ddot{y}_1 + c_1\dot{y}_1 + k_1\xi_1 = F_{m1} - F_{r1} - F_{c1} \\ M_2\ddot{y}_2 + c_2\dot{y}_2 + k_2\xi_2 = F_{m2} - F_{r2} + F_{c2} \end{cases}' \tag{1}$$

where $F_{ci} = f(y_1, y_2)$ is the coupling force, which is related to the displacements of motors $Y_1$ and $Y_2$, and $i$ = 1, 2. $F_{mi}$ is the thrust force, $F_{ri}$ is the friction, $M_i$ is the mass, $c_i$ is the viscosity coefficient, $k_i$ is the elasticity coefficient, $y_i$ is the displacement of the motor, and $\xi_i$ is the deformation between the mover and slider. The control method typically adopted by a dual-motor system is also shown in Figure 1. Each motor has an independent feedback controller (FBC); by designing the reasonable friction compensator (FC) and the synchronous control method, the synchronous motion performance of the dual-motor system is guaranteed.

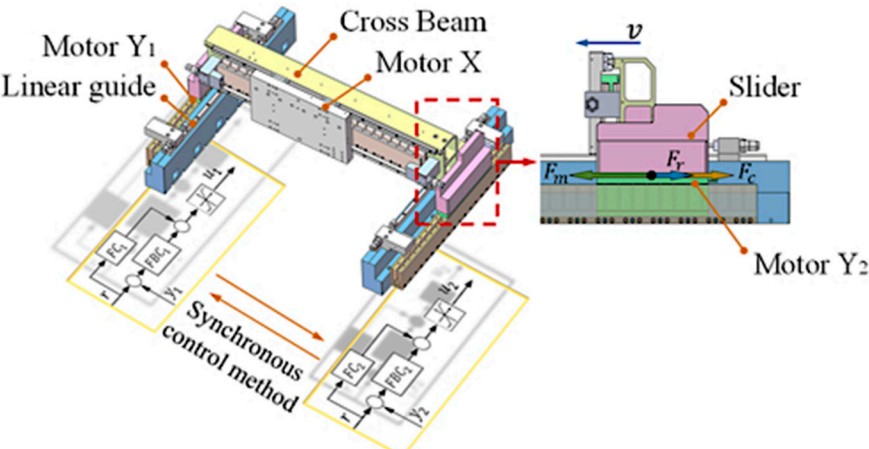

**Figure 1.** Schematic of the structure and control strategy of redundant direct-drive gantry platform.

The relationship between the motor thrust force and control system characteristics can be obtained as follows:

$$\begin{cases} F_{m1} = K_{u1}u_1 = K_{u1}sat\{u_{FC1} + u_{FBC1}\} \\ F_{m2} = K_{u2}u_2 = K_{u2}sat\{u_{FC2} + u_{FBC2}\} \end{cases},$$
(2)

where $K_{ui}$ is the thrust force constant of the motor, *sat* {} is the saturation function, and $u_{FCi}$ and $u_{FBCi}$ are the control voltages of FC and FBC, respectively. Because the coupling force is related to the displacements of motors $Y_1$ and $Y_2$, it appears with the generation of the synchronization error. When the synchronization error is eliminated, the influence of the coupling force on the motor movement will also be negligible, so the kinetic model can be re-expressed as follows:

$$\begin{cases} M_1\ddot{y}_1 + c_1\dot{y}_1 + k_1\xi_1 = F_{m1} \\ M_2\ddot{y}_2 + c_2\dot{y}_2 + k_2\xi_2 = F_{m2} \end{cases}.$$
(3)

We can establish the following transfer function of each motor system according to Equation (3):

$$g_{yi}(s) = \frac{b_{0i}s^{m_i} + b_{1i}s^{(m-1)_i} + \cdots + b_{mi}}{s^{n_i} + a_{1i}s^{(n-1)_i} + \cdots + a_{ni}},$$
(4)

where $a_{ni}$ and $b_{mi}$ are the parameters of the denominator and numerator polynomials, respectively, and $n_i$ and $m_i$ are the orders of the denominator and numerator polynomials, respectively. By designing a proper feedback controller $c_i(s)$, the transfer function between the input signal $r$ and output signal $y_i$ is as follows:

$$G_i(s) = \frac{y_i(s)}{r(s)} = \frac{c_i(s)g_{yi}(s)}{1 + c_i(s)g_{yi}(s)}.$$
(5)

From Equation (5), we can calculate the amplitude and phase characteristics of each closed-loop feedback control loop in the dual-motor system, and these characteristics determine the dynamic response characteristics of the motor system. Therefore, we can know when the dual-motor system is in a synchronous-state, and the motion of the motor system is mainly affected by its electric characteristic, structure, and control system, leading to the differences in the dynamic response characteristics between the motor systems.

### 2.2. Description of the Proposed Method

To make the dual-motor system have consistent dynamic response characteristics and realize the two motors' synchronous motion, we proposed a frequency-domain characteristic compensation-based adaptive fractional-order anti-saturation synchronous control method, which contains two modules: the FDC unification module (FDC-UM), which can adjust the dynamic response characteristics of the two motors to be consistent, and the adaptive fractional-order anti-saturation module (AFOAM), which can avoid the saturation effect of the control voltage adaptively. Figure 2 illustrates the working principle of the proposed synchronous control method. The FDC-UM performs the FDC unification through compensation in the feedback control loops. With the FDC-UM, the FDC of the dual-motor system can be unified. We have

$$\begin{cases} A_1 = A_2 = A_f \\ \varphi_1 = \varphi_2 = \varphi_f \end{cases},$$
(6)

where $A_1$, $A_2$ are the amplitude characteristics of the Equation (5), and $\varphi_1$, $\varphi_2$ are the phase characteristics. $A_f$, $\varphi_f$ are the amplitude and phase characteristics unified by the FDC-UM. In addition to the FDC-UM, the AFOAM is designed to prevent the peak voltage of the motion from reaching the voltage limitation. With these modules, the amplitude and phase characteristics are determined to be $A_a$ and $\varphi_a$ according to the input signal spectrum and theoretical calculation of the peak voltage. With the FDC unification and the elimination

of the nonlinear saturation effect, the dual-motor system will possess consistent dynamic response characteristics for the two motors and thus ensure their synchronous motion.

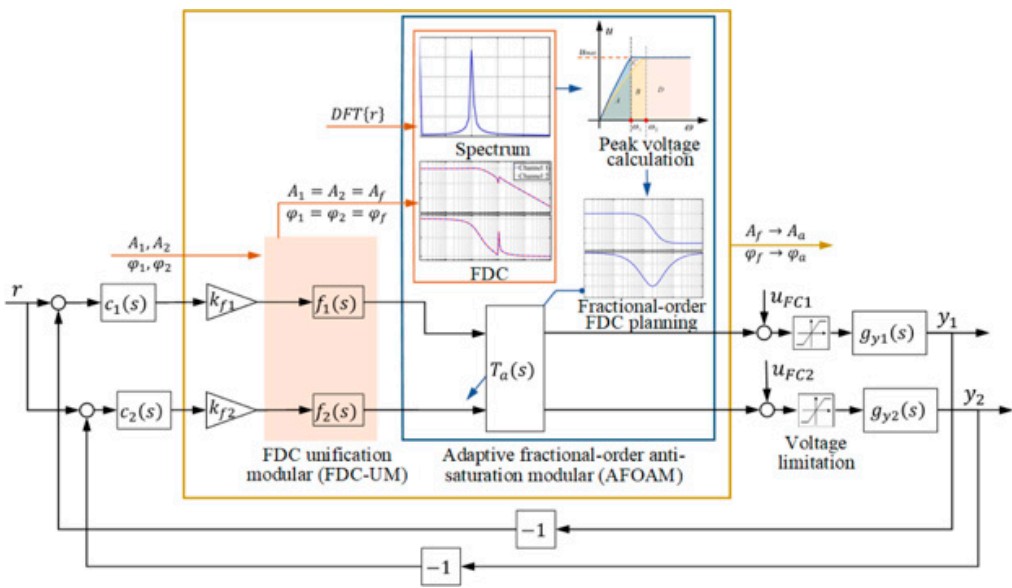

**Figure 2.** The transfer block diagram of the proposed adaptive fractional-order anti-saturation synchronous control method.

## 3. Adaptive Fractional-Order Anti-Saturation Synchronous Control Method

### 3.1. FDC Unification Module (FDC-UM)

According to the frequency-domain control theory, the dynamic response characteristics of the LTI system are determined by the FDC of the control system. In literature [28], the relationship between the input and output signals of the control system was analyzed in detail, and the output signal corresponding to the input signal $r = \sin(\omega_r t)$ was obtained as follows:

$$y = K \sin(\omega_r(t - \Delta t)). \tag{7}$$

The signals have the same frequency ($f = \omega_r/2\pi$), but there is a certain time delay $\Delta t_i$ and an amplitude ratio $K_i$ between them. Therefore, the synchronization error expression of the dual-motor system can be defined as follows:

$$e_{syn} = K_1 \sin(\omega_r(t - \Delta t_1)) - K_2 \sin(\omega_r(t - \Delta t_2)), \tag{8}$$

$$\Delta t_i = \varphi_i/\omega_r, \tag{9}$$

$$K_i = 10^{A_i/20}, \tag{10}$$

where $\varphi_i(\omega) = \arctan\left(\text{Im}\left[G_i(\omega)\right]/\text{Re}\left[G_i(\omega)\right]\right)$ and $A_i(\omega) = 20\log_{10}|G_i(\omega)|$ are the phase and amplitude characteristics of the transfer function shown in Equation (5), respectively. According to Equations (8)–(10), to achieve high-synchronous motion performance of the dual-motor system, both feedback control systems must have consistent FDC. That is, the dual-motor system must satisfy the following condition:

$$C(s)F(s)G_Y(s)K_F = 0, \tag{11}$$

where $C(s) = [c_1(s), c_2(s)]$, $G_Y(s) = \text{diag}(g_{y1}(s), g_{y2}(s))$. $F(s) = \text{diag}(f_1(s), f_2(s))$, in which $f_i(s)$ is the transfer function of the FDC-UM. $K_F = [k_{f1}, -k_{f2}]^T$ is the gain matrix used to adjust the

control performance of the system. According to Equations (5) and (11), we can derive the following when $\varphi_1 \geq \varphi_2$ in the main frequency domain.

$$\begin{cases} f_1(s) = \left(k_{f1}c_1(s)g_{y1}(s)\right)^{-1} k_{f2}c_2(s)g_{y2}(s) \\ f_2(s) = 1 \end{cases}, \tag{12}$$

When $\varphi_1 < \varphi_2$ in the main frequency domain,

$$\begin{cases} f_1(s) = 1 \\ f_2(s) = \left(k_{f2}c_2(s)g_{y2}(s)\right)^{-1} k_{f1}c_1(s)g_{y1}(s) \end{cases}. \tag{13}$$

After the compensation, the transfer function of the feedback control system is as follows:

$$G_{fi}(s) = \frac{y_i(s)}{r(s)} = \frac{k_{fi}f_i(s)c_i(s)g_{yi}(s)}{1 + k_{fi}f_i(s)c_i(s)g_{yi}(s)}. \tag{14}$$

Based on the analysis performed above, we know that the dynamic response characteristics of the control system are related to the frequency of the control signal. Through compensation, the dual-motor system meets the requirements of $A_{f1}(\omega) = A_{f2}(\omega)$ and $\varphi_{f1}(\omega) = \varphi_{f2}(\omega)$, so the two motor systems possess a unified dynamic response characteristic and thus can achieve a synchronous motion. To verify the effectiveness of the FDC-UM, we used the chirp signal for the experiments. Figure 3 shows the working principle of the FDC-UM and the synchronization error comparison by the control methods of the PID feedback control method (Method 1 in Figure 3a), the force compensation cross-coupling synchronous control method (Method 2 in Figure 3b), and the FDC-UM based synchronous control method (Method 3 in Figure 3c). Figure 3d shows that both Method 2 and Method 3 can ensure excellent synchronous-motion performance of the dual-motor system, and the FDC-UM-based synchronous control method achieves the best performance, which can avoid the effects of control signal variation in frequency.

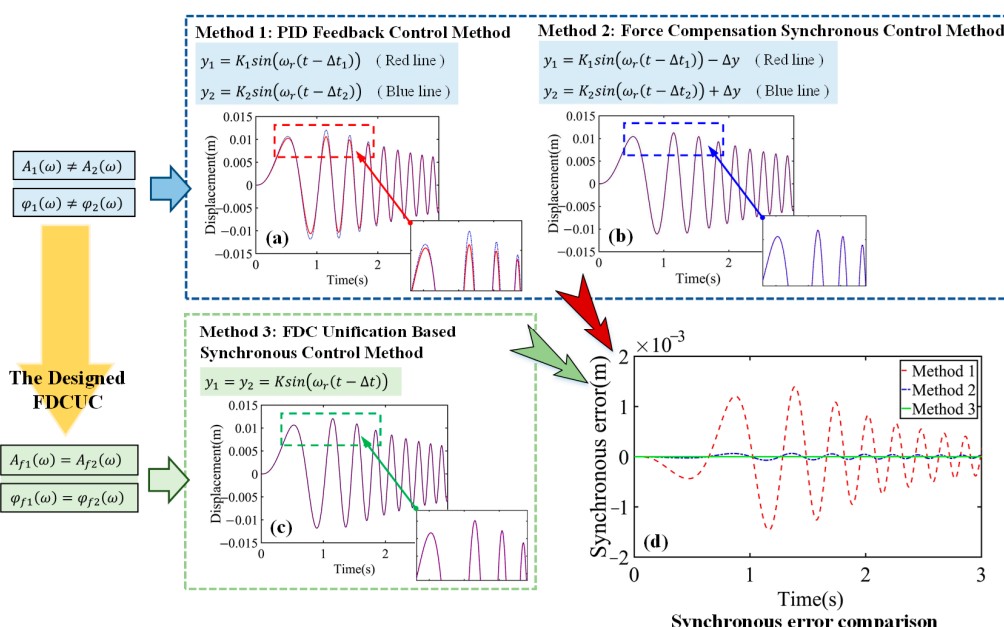

**Figure 3.** Working principle of the FDC-UM and comparison of synchronization errors with three different control methods: PID feedback method, force compensation method, and the proposed FDC-UM method.

Since the proposed FDC-UM is used for the LTI system, when the control voltage overpasses the voltage limitation, the control voltage will induce a nonlinear behavior and affect the dynamic response characteristics of the dual-motor system, as shown in Equation (15). When the total control voltage of the feedback control system $u_{FBC}$ and the friction compensator $u_{FC}$ are beyond the voltage limitation, the control voltage $u$ is limited to the highest (lowest) voltage $u_{max}$ ($-u_{max}$).

$$u = \begin{cases} -u_{\max}, & for \ (u_{FBC} + u_{FC}) \leq -u_{\max}, \\ u_{FBC} + u_{FC}, & \\ u_{\max}, & for \ (u_{FBC} + u_{FC}) \geq u_{\max}. \end{cases} \quad , \tag{15}$$

Figure 4 shows the influence of the saturation effect on the synchronization error of the dual-motor system. In Figure 4a, the synchronization errors of the dual-motor system are compared with the PID feedback control method and the FDC-UM-based synchronous control method. Figure 4b shows the control voltage of the motor system in the range of $[-u_{max}, u_{max}]$. We can see that the saturation effect influences the synchronization errors of the two control methods and deteriorates the performance of the FDC-UM. Therefore, to ensure the synchronous motion performance of the dual-motor system, the saturation effect needs to be eliminated.

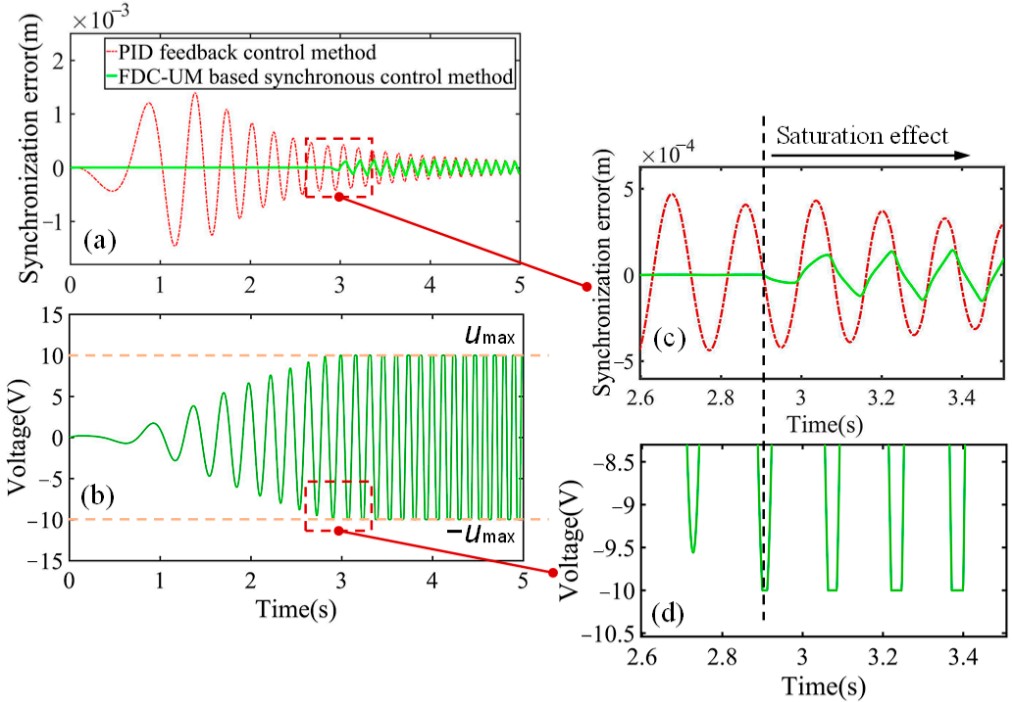

**Figure 4.** The saturation effect of the control voltage on the synchronization error of the dual-motor system: (**a**) under different control methods; (**b**) control voltage of the motor system in the range of $[-u_{max}, u_{max}]$; (**c**) partial enlarged view of (**a**); (**d**) partial enlarged view of (**b**).

### 3.2. Adaptive Fractional-Order Anti-Saturation Module (AFOAM)

3.2.1. Peak Voltage Calculation and Amplitude Characteristic Adjustment
Value Determination

To avoid the nonlinear problem caused by the saturation effect, it is necessary to recognize the determinants of the control voltage in the feedback control loop and calculate the required suppression value of the control voltage. According to the dynamic response

characteristics of the control system and Equations (4) and (14), the transfer function from the input signal $r$ to the feedback control voltage $u_{FBCi}$ is as follows:

$$G_{ui}(s) = \frac{u_{FBCi}(s)}{r(s)} = \frac{k_{fi}f_i(s)c_i(s)}{1 + k_{fi}f_i(s)c_i(s)g_{yi}(s)}. \tag{16}$$

In the frequency domain, $s = j\omega$, so the amplitude characteristic $A_{ui}(\omega)$ of Equation (16) is $20\log_{10}(|G_{ui}(\omega)|)$. Because the input signal amplitude in the frequency domain can be obtained by spectrum analysis, the calculation expression of the theoretical peak voltage of the control system can be expressed as

$$u_i^{\max} = |s_p(\omega_k)|10^{A_{ui}(\omega_k)/20} + u_{FCi}, \tag{17}$$

where $|s_p(\omega_k)|$ is the amplitude spectrum of the input signal when $\omega = \omega_k$, and $u_{FCi}$ is the friction compensation voltage. Therefore, the amount by which the theoretical peak voltage exceeds the voltage limitation can be calculated as follows:

$$\widetilde{u}_i = \text{sgn}_i(u)|u_i^{\max} - u_{\max}|, \tag{18}$$

in which the symbolic function for determining whether the theoretical peak voltage exceeds the voltage limitation is defined as follows:

$$\text{sgn}_i(u) = \begin{cases} 1, & \text{for } u_i^{\max} > u_{\max}, \\ 0, & \text{for } u_i^{\max} \leq u_{\max}. \end{cases} \tag{19}$$

According to Equation (17), we know that the control voltage can be changed by adjusting the spectrum of the input signal or the amplitude characteristic of the control system. Since the main work of this paper is the design of the synchronous control system, the adjustment of the input signal will directly affect the trajectory of the dual-motor system. Therefore, in order to avoid the saturation effect, the amplitude characteristic $A_{ui}(\omega)$ at $\omega_k$ must be adjusted to be the value as shown in Equation (20).

$$\overline{A}_{ui}(\omega_k) = 20\log_{10}\left|10^{A_{ui}(\omega_k)/20} - \widetilde{u}_i/|s_p(\omega_k)|\right|. \tag{20}$$

The amplitude characteristic adjustment value of each motor system is

$$\Delta A_i = \overline{A}_{ui}(\omega_k) - A_{ui}(\omega_k). \tag{21}$$

To ensure that the adjustment of the amplitude characteristic does not affect the unification of the dynamic response characteristics of the dual-motor system, it is necessary to unify the adjustment value to a smaller value $\Delta A$, so that the anti-saturation module can satisfy the adjustment requirements of both feedback control loops simultaneously. The amplitude characteristic adjustment value can be determined as follows:

$$\Delta A = -MAX\{|\Delta A_1|, |\Delta A_2|\}. \tag{22}$$

### 3.2.2. AFOAM Design and Implementation

A. AFOAM design

Combined with the above theoretical peak voltage calculation, we can obtain the saturation situation of the control voltage by analyzing the input signal when the frequency domain characteristics of the dual-motor system are determined. On this basis, we developed an AFOAM based on fractional-order frequency-domain control theory to realize the specific amplitude characteristic adjustment in which the saturation status of the control voltage is determined by analyzing the input signal when the frequency-domain characteristics of the dual-motor system are determined. The implementation block diagram of AFOAM is shown in Figure 5. When the input signal is determined, the controller

calculates the peak voltage according to Equation (17) to adjust the control system to avoid the saturation effect, i.e., not adjust the control parameters in real time during motion.

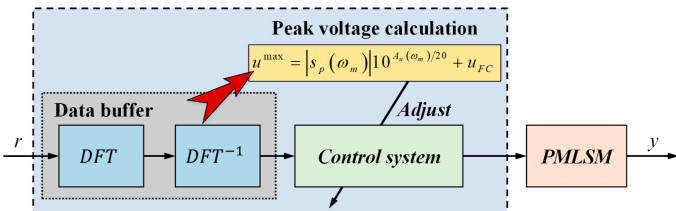

**Figure 5.** Implementation block diagram of AFOAM.

As shown in Equation (20), we only need to adjust the amplitude characteristic at $\omega_k$. Since the adjustment at a specific frequency will affect the amplitude and phase characteristics of the surrounding frequency and will further influence the dynamic response performance of the feedback control system, we propose the AFOAM in this paper to adjust the specific FDC accurately and flexibly by introducing the fractional-order lead-lag controller (FOLLC) as the fractional-order FDC planning target. The expression for the FOLLC is described as follows:

$$C_f(s) = K_C \left( \frac{1 + \lambda x s^\alpha}{1 + \lambda s^\alpha} \right), \tag{23}$$

where $K_C$, $\lambda$, and $x$ are coefficients. The order $\alpha$ can be any real number in the range (0, 2), which enables the FOLLC to accurately describe the specified FDC at the specified frequency through accurate parameter selection. The transfer function from the input signal $r$ to the feedback control voltage $u_{FBCi}$ with the AFOAM is as follows:

$$\overline{G}_{ui}(s) = \frac{k_{fi} f_i(s) c_i(s) T_a(s)}{1 + k_{fi} f_i(s) c_i(s) g_{yi}(s) T_a(s)}, \tag{24}$$

where $T_a(s)$ represents the transfer function of the AFOAM. With the AFOAM, the voltage amplitude characteristic of the feedback control system is changed as

$$\overline{A}_{ui}(\omega) = A_{ui}(\omega) + A_f(\omega), \tag{25}$$

where $A_f(\omega)$ is the amplitude characteristic of the FOLLC. According to the transfer block diagram of the proposed adaptive fractional-order anti-saturation synchronous control method, the transfer function of AFOAM can be expressed as follows:

$$T_a(s) = \frac{K_C(1 + \lambda x s^\alpha)}{(1 + \lambda s^\alpha - K_C(1 + \lambda x s^\alpha)) k_{fi} f_i(s) c_i(s) g_{yi}(s) + \lambda s^\alpha + 1}. \tag{26}$$

B. Parameter determination

It can be seen from Equation (26) that the parameters of the AFOAM can be determined by selecting the parameters of the FOLLC. Therefore, we first need to establish the relationship between the parameters of the FOLLC and the FDC adjustment. Equation (23) can be rewritten as

$$C_f(j\omega) = K_C \left( \frac{1 + x\lambda(j\omega)^\alpha}{1 + \lambda(j\omega)^\alpha} \right). \tag{27}$$

The amplitude and phase characteristics of the FOLLC can be calculated as follows:

$$A_f(\omega) = 20 \log_{10} \left( K_C \sqrt{\frac{1 + x^2 \lambda^2 \omega^{2\alpha} + 2x\lambda\omega^\alpha \cos(\alpha\pi/2)}{1 + \lambda^2 \omega^{2\alpha} + 2\lambda\omega^\alpha \cos(\alpha\pi/2)}} \right), \tag{28}$$

$$\varphi_f(\omega) = \tan^{-1}\left[\frac{(x-1)\lambda\omega^\alpha \sin(\alpha\pi/2)}{1 + (x+1)x\lambda^2\omega^{2\alpha}\cos(\alpha\pi/2)}\right], \tag{29}$$

Let

$$g(\omega) = K_C\sqrt{\frac{1 + x^2\lambda^2\omega^{2\alpha} + 2x\lambda\omega^\alpha \cos(\alpha\pi/2)}{1 + \lambda^2\omega^{2\alpha} + 2\lambda\omega^\alpha \cos(\alpha\pi/2)}}. \tag{30}$$

The logarithmic function is monotonic, and thus, $g(\omega)$ represents the trend of $A_f$ $(\omega)$. Because the FOLLC has equal-order properties, that is, the orders of the differential operator in the numerator and denominator polynomials are the same, we can deduce that the FOLLC has the following properties: (1) $A_f(\omega)|_{\omega\to0} = 20\log_{10}(K_C)$ and $A_f(\omega)|_{\omega\to+\infty} = 20\log10(K_C x)$; (2) $\varphi_f(\omega)|_{\omega\to0} = \varphi_f(\omega)|_{\omega\to+\infty} = 0$; (3) when $\omega = \omega_m$, the FOLLC reaches its extreme phase characteristic value. By deriving Equation (29), we obtain

$$\omega_m = \left(\lambda\sqrt{x}\right)^{-1/\alpha}, \tag{31}$$

The parameter $\lambda$ can be obtained by Equation (32).

$$\lambda = \left(\sqrt{x}\omega_m^\alpha\right)^{-1}, \tag{32}$$

Because the middle- and low-frequency domains are the main control-frequency domains of the motor system, the FDC adjustment should avoid excessive influence on the amplitude and phase characteristics in these areas. However, owing to the continuity of the FDC adjustment, this influence cannot be completely avoided. To minimize the influence of the FDC adjustment caused by the AFOAM on the outer area of the specified angular frequency, especially the middle- and low-frequency domains, we need to let $K_C = 1$. According to the above analysis, to minimize the influence range of the FDC adjustment, we determine $A_f(\omega)|_{\omega\to+\infty} = \Delta A$ and $|A_f(\omega_k) + \Delta A| \le \varepsilon_1$. Because the logarithmic function is a monotonic function, $A_f(\omega_k)$ satisfies $|\dot{g}(\omega_k)| \le \varepsilon_2$ simultaneously, where $\varepsilon_1$ and $\varepsilon_2$ represent the minimum values. Thus, parameter $x$ can be calculated by Equation (33).

$$x = 10^{\Delta A/20}, \tag{33}$$

To simplify the calculation, we let $A_f(\omega_k) \approx \Delta A$ and substitute it into Equation (28), and then we can obtain the relationship between $\omega_j$ and $\alpha$ by the following expression:

$$\omega_j = \omega_k\left(-2\sqrt{x}\cos(\alpha\pi/2)/(x+1)\right)^{2/\alpha}, \tag{34}$$

where $\omega_j$ satisfies the condition $\omega_m = (\omega_k\omega_j)^{1/2}$, and $(\omega_k-\omega_j)$ is the range of the FDC adjustment. Since the variables $x$, $\omega_k$, and $\omega_j$ are not less than zero, the order $\alpha$ should be in the range of [1, 2). We can see that when the order $\alpha = 1$, $\omega_j = 0$, which means that the FDC adjustment of the AFOAM will influence the amplitude and phase characteristics at all frequencies. Under this condition, because $\omega_m = (\omega_k\omega_j)^{1/2}$, $\omega_m$ will be zero by calculation, so the parameters of the AFOAM cannot be determined uniquely and adaptively. Therefore, to reduce the influence of the FDC adjustment, the order must be in the range of (1, 2).

Through the derivation of Equation (30), we obtain

$$\dot{g}(\omega_k) = \frac{\left(1 + \frac{\omega_k^\alpha}{\omega_j^\alpha} + 2\frac{\omega_k^{\alpha/2}}{\sqrt{x}\omega_j^{\alpha/2}}\cos(\alpha\pi/2)\right)^{1/2}\left((x-1)\alpha\frac{\omega_k^{(\alpha/2-1)}}{\sqrt{x}\omega_j^{\alpha/2}}\cos(\alpha\pi/2) + \frac{\omega_k^{\alpha/2}}{\sqrt{x}\omega_j^{\alpha/2}} + \sqrt{x}\frac{\omega_k^{\alpha/2}}{\omega_j^{\alpha/2}} + \frac{\omega_k^\alpha}{\omega_j^\alpha}\cos(\alpha\pi/2)\right)}{\left(1 + x\frac{\omega_k^\alpha}{\omega_j^\alpha} + 2\sqrt{x}\frac{\omega_k^{\alpha/2}}{\omega_j^{\alpha/2}}\cos(\alpha\pi/2)\right)^{1/2}\left(1 + \frac{\omega_k^\alpha}{x\omega_j^\alpha} + 2\frac{\omega_k^{\alpha/2}}{\sqrt{x}\omega_j^{\alpha/2}}\cos(\alpha\pi/2)\right)^2}, \tag{35}$$

For Equation (35), $|\dot{g}(\omega_k)| \le \varepsilon_2$, so there is a minimum value $\varepsilon$, making $|\dot{g}(\omega_k) - \varepsilon| = 0$. Combining Equations (34) and (35), and the above conditions, the optimal model for parameter $\alpha$ can be deduced as follows:

$$\begin{cases} \min Z(\alpha) = \omega_k - \omega_k\left(-2\sqrt{x}\cos(\alpha\pi/2)/(x+1)\right)^{2/\alpha} \\ s.t. \quad 1 < \alpha < 2. \\ \quad \left|A_f(\omega_k) + \Delta A\right| \leq \varepsilon_1. \\ \quad \left|\dot{g}(\omega_k) - \varepsilon\right| = 0. \end{cases} \qquad (36)$$

Through Equation (36), parameter $\alpha$ can be solved and determined. Then, parameters $\lambda$ and $\omega_m$ can be calculated by combining Equations (31)–(33), and $\omega_m = (\omega_k\omega_j)^{1/2}$. Therefore, through the establishment of the relationship between the FOLLC and FDC adjustment, the parameters $K$, $x$, $\lambda$, and $\alpha$ of the AFOAM can be determined properly.

## 4. Experiment

### 4.1. Experimental System

Based on the above theoretical analysis and derivation, we implemented the proposed adaptive anti-saturation synchronous control system using a dSPACE prototyping system and conducted experiments on the redundant direct-drive gantry platform to verify the improvement in the synchronous motion performance of the proposed method. Figure 6 shows the experimental setup. The Y-direction motion of the redundant direct-drive gantry platform was driven by AUM3 PMLSMs from Akribis. Each motor was equipped with RGS20-S grating from Renishaw, with a resolution of 0.1 μm.

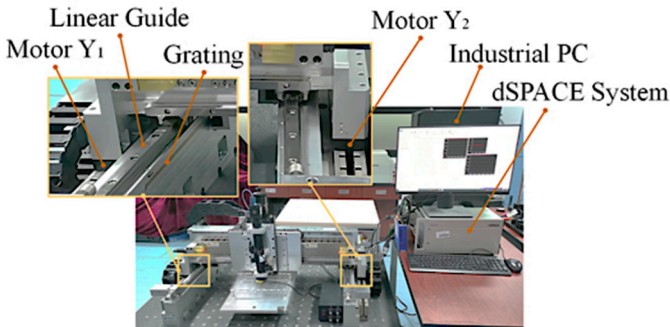

**Figure 6.** Experimental system.

The beam had a length of 836 mm, and its weight exceeded 25 kg, which caused the synchronous motion performance of the dual-motor system to be significantly affected by the synchronous control method. During the experiments, the dSPACE system controlled the PMLSMs in real time, and the feedback signals were provided by the gratings. By establishing the mathematical model of the motor system and using a frequency sweep experiment to identify the model parameters, the transfer functions of the motor systems were calculated as follows:

$$\begin{cases} g_{y1}(s) = \dfrac{0.9183s^3 + 5.994s^2 + 8959s + 8499}{s^5 + 16.92s^4 + 1.061e04s^3 + 7.916e04s^2 + 5.448e04s} \\ g_{y2}(s) = \dfrac{0.8818s^3 + 8.583s^2 + 8918s + 2.223e04}{s^5 + 12.51s^4 + 1.104e04s^3 + 3.237e04s^2 + 4.215e04s} \end{cases} \qquad (37)$$

The dual-motor system adopted a proportional–integral–derivative (PID) feedback control method. According to the controlled object models shown in Equation (37), the Ziegler–Nichols tuning rules were used to obtain a set of PID parameters that were more consistent with the synchronous-motion requirements of the dual-motor system. The PID controller and its parameters are as follows:

$$\begin{cases} c_1(s) = \left(27.41s^2 + 388.52s + 1.05\right)/s \\ c_2(s) = \left(28.12s^2 + 275.44s + 1.12\right)/s \end{cases} \qquad (38)$$

According to Equations (37) and (38), the FDC diagrams of the dual-motor system under the open-loop, closed-loop, and control methods in this study were obtained, as shown in Figure 7. As seen in the figure, even if the FDC values of the motor systems are significantly different, the closed-loop FDC of the motor systems can be guaranteed to be relatively consistent through the adjustment of the PID feedback controllers. However, there is still a large difference in the amplitude and phase characteristics around $\omega = 1$ rad/s. Because the phase characteristic of Motor $Y_2$ is ahead of that of Motor $Y_1$, the FDC-UM can be calculated using Equation (13). When $K_F = [1, -1]^T$, the FDC-UM is given by Equation (39). With the FDC-UM, the FDC of the two motor systems can be made consistent, as shown in Figure 7c.

$$
\begin{cases}
f_{11}(s) = 1 \\
f_{22}(s) = \dfrac{\begin{aligned}&25.17s^{11} + 836s^{10} + 5.323e05s^9 + 1.33e07s^8 + \\ &2.804e09s^7 + 4.909e10s + 1.672e11s^5 + 2.638e11s^4 + \\ &1.399e11s^3 + 3.761e08s^2\end{aligned}}{\begin{aligned}&24.8s^{11} + 903.8s^{10} + 5.244e05s^9 + 1.447e07s^8 + \\ &2.784e09s^7 + 5.286e10s^6 + 3.228e11s^5 + 6.536e11s^4 + \\ &3.361e11s^3 + 1.356e09s^2\end{aligned}}
\end{cases}
, \tag{39}
$$

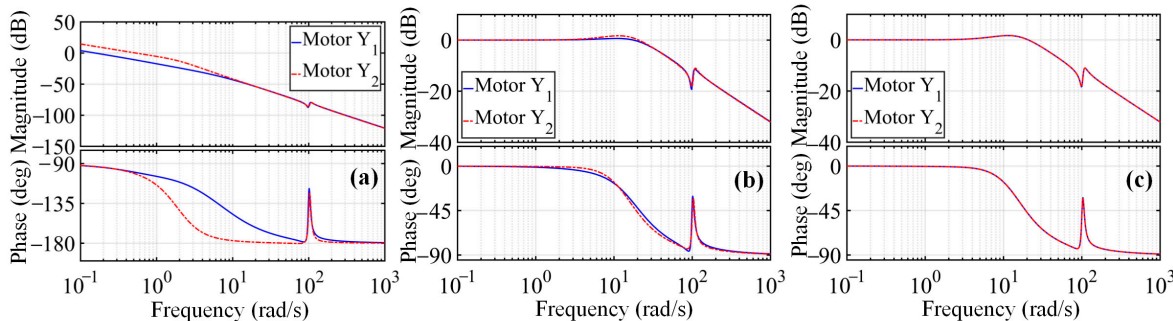

**Figure 7.** Comparison of the dual-motor system FDC: (**a**) open-loop control; (**b**) closed-loop control; (**c**) closed-loop control with the designed FDC-UM.

According to Equation (17), it is necessary to obtain the friction compensation voltage to accurately calculate the adaptive gain adjustment value. The Stribeck friction model is a commonly used friction model that can more comprehensively reflect the change in the friction of the guide when the PMLSM moves continuously [29]. Based on the Stribeck friction model, the functions of the friction compensation voltage are shown in Equations (40) and (41), and the curves are shown in Figure 8. From Figure 8, we can see the high nonlinearity and complexity of friction, and there is a sudden change in friction during the low-speed period, which seriously deteriorates the accuracy of the motor system. However, when the speed reaches a certain value, the friction remains within a certain range. At this time, the friction changes very little; therefore, the friction compensation voltage used in the calculation of Equation (17) can be clearly obtained.

$$
u_{FC1} = \begin{cases}
\left(11.13 + 0.00037v_1 + 5.51e^{-(v_1/0.0123)^2}\right)/72, \ for \ v_1 > 0, \\
\left(-13.15 + 0.00011v_1 - 5.33e^{-(v_1/0.0139)^2}\right)/72, \ for \ v_1 < 0.
\end{cases}
, \tag{40}
$$

$$
u_{FC2} = \begin{cases}
\left(13.03 + 0.00047v_2 + 5.25e^{-(v_2/0.0103)^2}\right)/72, \ for \ v_2 > 0 \\
\left(-16.22 + 0.00018v_2 - 5.51e^{-(v_2/0.0156)^2}\right)/72, \ for \ v_2 < 0.
\end{cases}
\tag{41}
$$

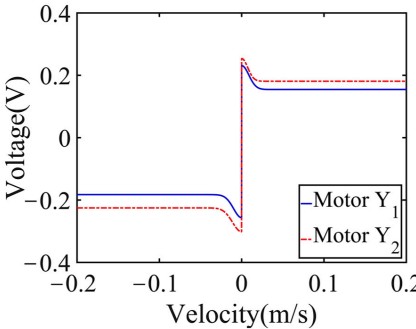

**Figure 8.** Diagram of the friction compensation voltage.

*4.2. Synchronized Motion Experiments*

4.2.1. Verification of Peak Voltage Calculation

The AFOAM calculates the theoretical peak voltage based on the spectrum of the input signal and designs the corresponding fractional-order FDC, aiming to avoid the saturation effect of the control voltage. Therefore, the correctness of the calculation expression for the theoretical peak voltage directly affects the effectiveness of the proposed method in improving the synchronization accuracy. The experiments verified the accuracy of the calculation under different accelerations and velocities.

To accurately analyze the spectrum of the input signal, we modified the short-term input signal into a long-period signal through periodic processing and calculated the spectrum according to the discrete Fourier transform shown in Equation (42).

$$s_p(\omega) = T_s^2 \sum_{n=-\infty}^{+\infty} \left( \sum_{k=-\infty}^{n} \left( \sum_{i=-\infty}^{k} a(i) \right) \right) e^{-j\omega n}, \tag{42}$$

where $s_p(\omega)$ is the spectrum of the input signal, $T_s$ is the sampling time, $a(i)$ denotes the discrete acceleration sequence, $k$ is the number of discrete acceleration sequences, and n is the number of discrete velocity sequences. The cubic curve shown in Equation (43) is a commonly used point-to-point motion planning that can realize the adjustment of acceleration, velocity, and displacement. Therefore, this curve was used for experimental analysis. Figure 9 shows the motion planning when the displacement is 0.05 m, maximum speed is 0.4 m/s, and maximum acceleration is 30 m/s$^2$.

$$r = \begin{cases} a_0 + a_1 t + a_2 t^2 + a_3 t^3, \, for \, t \leq t_a, \\ a_0 + a_1 t_a + a_2 t_a t + a_3 t_a^3, for \, t_a < t \leq (t_a + t_c), \\ a_0 + a_1(t - t_c) + a_2 t_a t_c + a_2(t - t_c)^2 + a_3(t - t_c)^3, \\ for \, (t_a + t_c) < t \leq (2t_a + t_c). \end{cases} \tag{43}$$

where $a_i$ (i = 1, 2, ... , k) is a parameter of the cubic curve, $t_a$ is the acceleration time, $t_c$ is the uniform motion time, and r is the displacement.

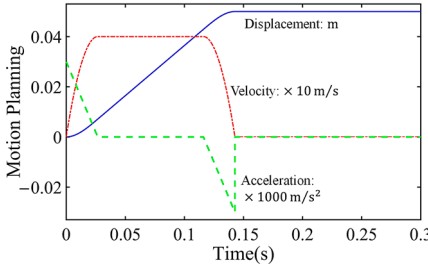

**Figure 9.** Motion planning of a cubic curve.

Figure 10 shows the calculated amplitude spectrum of the motion planning with different accelerations under the same acceleration time, and Figure 11 shows the calculated amplitude spectrum of the motion planning with different velocities under the same acceleration. It can be seen from Figure 10 that, as the acceleration increases, the amplitude spectrum at the non-zero frequency increases, but all accelerations are at a similar frequency. The results in Figure 11 show that, under the same acceleration, with an increase in velocity, the amplitude spectrum at the non-zero frequency increases, but the frequencies are all retained within a small range.

Based on the spectrum analysis of the input signal, the voltage amplitude characteristics of the motor system at the corresponding angular frequency were calculated. Table 1 shows the comparison results of the theoretical and experimental peak voltages under different motion planning conditions. The relative error $\delta$ in the table is calculated as follows:

$$\delta = |u_{cal} - u| / u \times 100\%, \tag{44}$$

where $u$ is the experimental peak voltage, and $u_{cal}$ is the calculated theoretical peak voltage. In the experimental results shown in Table 1, the maximum relative error between the experimental and the calculated theoretical peak voltages was 3.86%, which proves the accuracy of the calculation of the theoretical peak voltage; thus, Equation (17) can guarantee the accurate fractional-order FDC planning for the adaptive fractional-order anti-saturation synchronous control method.

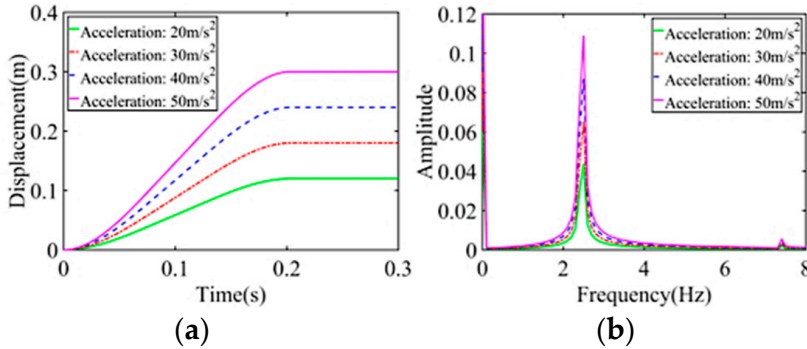

**Figure 10.** Calculated amplitude spectrum of different accelerations under the same acceleration time (80 ms): (**a**) displacement comparison; (**b**) amplitude spectrum comparison.

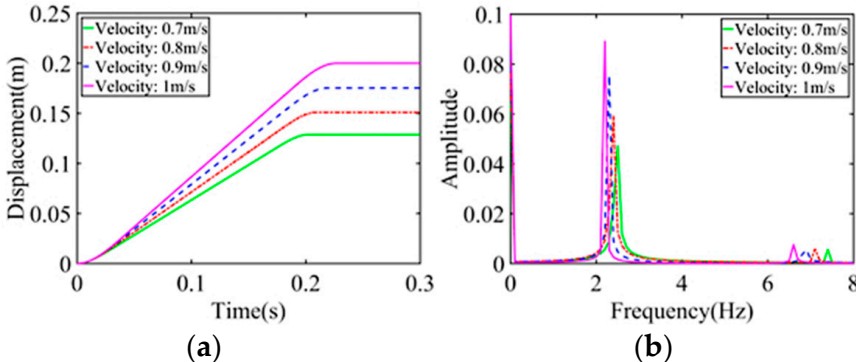

**Figure 11.** Calculated amplitude spectrum of different velocities under the same acceleration (50 m/s$^2$): (**a**) displacement comparison; (**b**) amplitude spectrum comparison.

**Table 1.** Comparison of calculated theoretical and experimental peak voltages.

| Motion Planning | | Amplitude Spectrum | | Voltage Amplitude Characteristic of the Motor $Y_1$ (dB) | Peak Voltage (V) | | $\delta$ |
|---|---|---|---|---|---|---|---|
| Acceleration (m/s²) | Velocity (m/s) | Frequency (Hz) | Peak Value | | Experiment | Calculation | |
| 20 | 0.8 | 2.5 | 0.0435 | 50.3265 | 14.8562 | 14.2828 | 3.86% |
| 30 | 1.2 | 2.5 | 0.0653 | 50.3265 | 22.2843 | 21.4406 | 3.79% |
| 40 | 1.6 | 2.5 | 0.0870 | 50.3265 | 29.7124 | 28.5655 | 3.86% |
| 50 | 2.0 | 2.5 | 0.1088 | 50.3265 | 37.1405 | 35.7233 | 3.82% |

### 4.2.2. Synchronization Accuracy Verification

To further verify the correctness of the calculation for the proposed adaptive fractional-order anti-saturation synchronous control method and its effectiveness in improving the synchronization accuracy of the dual-motor system, the cubic curve shown in Equation (43) was selected to perform the high-speed high-acceleration motion experiments. According to the above analysis and experiments of the FDC-UM, the parameters of the PID feedback controller selected in the experiments are shown in Equation (38), and those of the friction compensator are shown in Equations (40) and (41). When the dual-motor system was controlled by the proposed synchronous control method, the AFOAM analyzed the spectrum of the input signal and the FDC of the control system and then used Equation (36) to calculate the order of the FOLLC as 1.3190. According to Equations (31)–(33), $\omega_j = 7.1740$, $K_C = 1$, $x = 0.8055$, and $\lambda = 0.1381$. Therefore, using Equation (26), the transfer function of the AFOAM was calculated as Equation (45). According to Equation (45), we could draw the adjustment curve of the voltage amplitude characteristic realized by the AFO-AM, as shown in Figure 12a. An FDC comparison of the motor system with and without the designed AFO-AM is shown in Figure 12b.

The anti-saturation module can avoid synchronization errors owing to the voltage limitation; therefore, compared with the force compensation cross-coupling synchronous control method, the proposed synchronous control method can avoid the artificial secondary parameter adjustment of the control system to achieve high-synchronization accuracy under different motion plannings. Figure 13 shows the experimental results of the synchronous motion of the dual-motor system. Under force compensation cross-coupling synchronous control, both motor systems were affected by long-term saturation, which led to a large synchronization error. However, there was no control voltage saturation under the proposed synchronous control, as shown in Figure 13b, so the synchronization error was reduced obviously, as shown in Figure 13c. Table 2 shows the comparison results of experimental data under three different motion plannings, in which we use the commonly used indicators, such as absolute maximum value (|MAX|), mean absolute error (MAE), and root mean square error (RMSE) to compare the synchronization error under different control methods. Under the proposed adaptive fractional-order anti-saturation synchronous control, the root-mean-square synchronization error of the dual-motor system can be reduced by more than 77.580% compared with that under the force compensation synchronous control. The comparison results of the three sets of experimental data verify that the proposed adaptive fractional-order anti-saturation synchronous control method has obvious advantages in improving the synchronization accuracy of the dual-motor system.

$$T_a(s) = \frac{1 + 0.1112s^{1.3190}}{0.0269s^{1.3190}\left(\dfrac{25.17s^5 + 521.1s^4 + 2.479e05s^3 + 3.714e06s^2 + 3.311e06s + 8924}{s^6 + 16.92s^5 + 10610s^4 + 79160s^3 + 54480s^2}\right) + 0.1381s^{1.3190} + 1}. \tag{45}$$

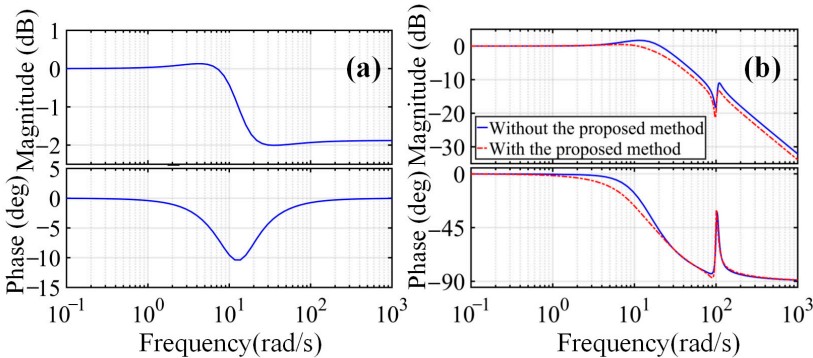

**Figure 12.** FDC adjustment of the proposed adaptive fractional-order anti-saturation method: (**a**) fractional-order FDC planning target for the voltage amplitude characteristic; (**b**) FDC comparison of the two motor systems with and without the proposed method.

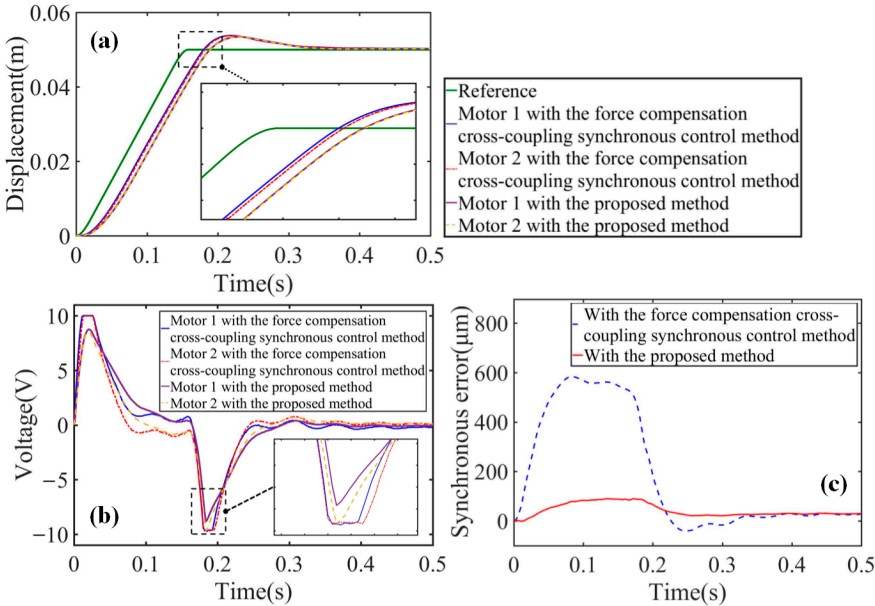

**Figure 13.** Synchronous motion experimental results of the dual-motor system under different synchronous control methods: (**a**) displacement comparison; (**b**) control voltage comparison ($|u_{max}| = 10$V); (**c**) synchronization error comparison.

**Table 2.** Experimental data comparison results of the synchronous motion experiments.

| Experiment | Motion Planning | | | Control Method | Synchronization Error (μm) | | | RMSE Reduction (%) |
|---|---|---|---|---|---|---|---|---|
| | Displacement (m) | Velocity (m/s) | Acceleration (m/s²) | | \|MAX\| | MAE | RMSE | |
| 1 | 0.05 | 0.35 | 30 | Method 2 | 584.500 | 117.259 | 221.443 | - |
| | | | | Proposed method | 89.800 | 36.987 | 42.132 | 80.974 |
| 2 | 0.05 | 0.5 | 30 | Method 2 | 561.600 | 80.388 | 161.193 | - |
| | | | | Proposed method | 52.900 | 31.436 | 32.338 | 79.938 |
| 3 | 0.05 | 0.5 | 50 | Method 2 | 769.000 | 106.681 | 220.616 | - |
| | | | | Proposed method | 141.400 | 40.019 | 49.454 | 77.580 |

**Method 2:** Force compensation cross-coupling synchronous control method.

## 5. Discussion

Synchronization error of a dual-motor system will usually lead to degradation of equipment performance. Several researchers focused on the cross-coupling synchronous

control method, by reducing the synchronization error from the aspects of the compensation method and the control parameter tunning method, to improve the synchronous motion performance of a dual-motor system [19,30]. However, the cross-coupling synchronous control method is usually used to compensate the control voltage after the synchronization error occurred, so there is a time delay between the generation and the compensation of the synchronization error. As shown in Figure 3, under the control of the chirp signal, although the cross-coupling synchronous control method could greatly reduce the synchronization error of the dual-motor system, it was difficult to eliminate the synchronization error completely due to the constant changes in the amplitude and frequency of the input signal and the inconsistent response characteristics of the dual-motor system. The experimental results showed the relationship between the dynamic response characteristics and the FDC of the motor systems. With the proposed adaptive fractional-order anti-saturation synchronous control method, the FDC of the dual-motor system was adjusted to be consistent to guarantee the same dynamic response output of the dual-motor system. Compared with the cross-coupling synchronous control method, our method can suppress the generation of the synchronization errors of the dual-motor system and eliminate the influence of the saturation effect and thus can achieve a better synchronous-motion control performance, as shown in Table 2. It can be seen from the experimental results in Figure 13 that our method can effectively avoid the non-linear phenomenon of control voltage saturation and ensure the high-precision synchronization accuracy of the dual-motor system. Considering the problem of the hysteresis nonlinearity and gap nonlinearity of the dual-motor system, the synchronous control performance of the proposed method may be affected. Next, we will tackle this problem to further improve the synchronous control method for the dual-motor system.

## 6. Conclusions

In this study, we proposed an adaptive fractional-order anti-saturation synchronous control method to deal with the asynchronous problem of the dual-motor system. With the elimination of the differences in FDC by the designed FDC-UM, both motor systems possessed consistent dynamic response characteristics. Based on the theoretical peak voltage calculation, we further developed an adaptive fractional-order anti-saturation module (AFOAM) to eliminate the nonlinear saturation effect caused by the control voltage saturation and thus effectively improved the synchronization accuracy of the dual-motor system. The experimental results showed that the maximum relative error between the calculated and experimental peak voltage was only 3.86%, which confirmed the correctness of the theoretical calculation of the peak voltage in this paper. On this basis, the proposed adaptive fractional-order anti-saturation synchronous control method could prevent the control voltage from reaching the limitation and reduce the root-mean-square synchronization error from 221.443 μm to 42.132 μm, with a reduction of 80.974%. The experimental results demonstrated that the proposed method can effectively suppress the inconsistent characteristics of the dual-motor system and avoid the nonlinear saturation effect. Therefore, the proposed method can effectively improve the synchronization accuracy of the dual-motor system.

**Author Contributions:** Writing–original draft, Y.Z.; Writing—review & editing, J.G. and L.Z. All authors have read and agreed to the published version of the manuscript.

**Funding:** This work was supported by the National Natural Science Foundation of China under Grant No. 52075106, No. 51905108, and No. U20A6004.

**Conflicts of Interest:** The authors declare no conflict of interest.

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
