# Peer review of "Adaptive Fractional-Order Anti-Saturation Synchronous Control for Dual-Motor Systems"

_applsci, doi:10.3390/app13042307_

Round 1
Reviewer 1 Report
1) Provide a list of contributions and give sufficient evidences to support your claim.
2) The necessity of using a fractional-order scheme has not been motivated well. What are the features of the problem that make the fractional-order tools more appropriate?
3) Literature review is week. There are recent publications addressing constraints in fractional-order systems that could be used. Consider improving the literature review by discussing the following publications:
[R1]. "Model Predictive Control of Fractional Order Systems", 2014. [https://doi.org/10.1115/1.4026493]
[R2]. "Constrained Control of Semilinear Fractional-Order Systems: Application in Drug Delivery Systems", 2020. [http://doi.org/10.1109/CCTA41146.2020.9206328]
[R3]. "Null Controllability of Fractional Dynamical Systems with Constrained Control", 2017. [https://doi.org/10.1515/fca-2017-0029]
4) I failed to understand why the proposed method is adaptive. Which parameter is adapting our time? What is the adaptation law?
5) In Equation (35), if w_j approaches zero, the expression becomes indefinite. How can one ensure that w_j will never approach zero?
6) The manuscript lacks a comparison study with a state-of-the-art method? Is it possible to carry out such a comparison study? That would help the readers to understands upsides and downsides of the proposed method.
Author Response
Thank you very much for reviewing this article and pointing out the problems. Please see the attachment!

Reviewer 2 Report
This manuscript addresses a challenging technical problem. But in order to merit publication, several questions have to be solved:
1) The terms k_1*y_1 and k_2*y_2 of expressions (1) and (3) have to be justified. What elasticity do represent k_1 and k_2?.
2) Which is the expression of the coupling force?, does it depend of the difference between y_1 and y_2?. Is elastic the beam?
3) What is the FDC-UM?. It appears in line 177 and is not defined. Saying in the 3.1 Subsection title that is "FDC unification modular" is not a definition.
4) It is very difficult to distinguish plots in figures 3 and 12.
5) The meaning of "adaptive" is unclear here: is the correction of the amplitude characteristic at a given frequency done in real time or is precomputed based on the spectrum of a trajectory to be carried out?.
6) In the case of carrying out controller changes in real time, asymptotic stability has to be proved. In the case that it is precomputed, how do you handle saturation because of real time disturbances?.
7) In fact, you say in the Discussions (line 464) that "the constant changes in amplitude and frequency of the input signal .....". How do you handle this in your method?.
8) A justification is required of why a fractional order compensator is used. Could not you get a similar result with an integer order controller with a more complex structure than a PID?. At the end, you must convert the fractional transfer function into a high degree integer order transfer function, in order to implement this. Then, why no directly design an integer order transfer function?.
9) The proposed antisaturation method has to be justified. Why do not use an antiwindup term to alleviate saturation effects?.
Author Response

(The authors gave the same response as above.)

Round 2
Reviewer 1 Report
No further comment.
Author Response
Many thanks for the reviewer’s affirmation, which is a great incentive for us.
Reviewer 2 Report
Some of my concerns have been answered but other still require a cogent explanation:
1) If k_1 and k_2 are stiffness coefficient, they must depend on incremental variables and not on absolute variables y_1 and y_2. Moreover I do not see the physical phenomenon that justifies these terms (I can only understand the coupling force that would depend on the difference between y_1 and y_2).
2) In line 112 of the previous version, only a generic f(y_1,y_2) is given, in the new version there is not a expression in line 112. The expression of the coupling force is still missing.
3) Considering real time disturbances is something fundamental in the design of a control system. And this is especially true in your approach, in which you base your control system in a previous computing of the controller in order to avoid saturation, which should be obviously based on an accurate model without disturbances. Then this issue must be addressed.
4) My question about the need of using fractional controllers has not been solved yet since though you can get a good fitting with your controller, its implementation is an integer high order transfer function. You must justify that a fractional controller works better than an integer order controller of higher orders than a PID by carrying out a comparative study, at least in the frequency domain, of shaping the desired frequency response with your controller and other integer order controllers of, for example, third or fourth order.
Author Response
We are very grateful for the reviewer’s comments! Please see the attachment!

Round 3
Reviewer 2 Report
My concerns have been adequately addressed. The paper can be published.